# Experimental Study on Vibration Velocity of Piled Raft Supported Embankment and Foundation for Ballastless High Speed Railway

**Qiang Fu** , **Meixiang Gu \*** , **Jie Yuan and Yifeng Lin**

School of Civil Engineering, Guangzhou University, Guangzhou Higher Education Mega Center, 230 Wai Huan Xi Road, Guangzhou 510006, China
\* Correspondence: mxgu@gzhu.edu.cn

**Abstract:** In recent years, the high development of high-speed railway lines cross through areas with poor geological conditions, such as soft soil, offshore and low-lying marsh areas, resulting geotechnical problems, such as large settlements and reduction of bearing capacity. As a new soil reinforcement method in high speed railway lines, the piled raft structure has been used to improve soil conditions and control excess settlement. In order to study the dynamic behavior of piled raft supported ballastless track system in soft soil, an experimental study on vibration velocities of piled raft supported embankment and foundations is presented in soft soil with different underground water levels. Vibration velocities at specified positions of the piled raft supported embankment and foundations are obtained and discussed. The vibration velocity curves on various testing locations of piled raft foundations are clearly visible and have sharp impulse and relaxation pattern, corresponding to loading from train wheels, bogies, and passages. Vibration velocity distribution in the horizontal direction at three train speeds clearly follows an exponential curves. Most of the power spectrums of vibration velocity at various locations are mainly concentrated at harmonic frequencies. The change in water level has slight impaction on the peak spectrum of vibration velocity at harmonic frequencies. The vibration power induced by train loads are transmitted, absorbed, and weakened to a certain extent through embankment and piled raft structure. The dynamic response character of embankments are affected by their self-vibration characteristics and the dynamic bearing capacity of the piled raft structure.

**Keywords:** model test; piled raft; high speed railway; vibration velocity; ground water level

## 1. Introduction

High-speed moving train loads produce large amounts of vibration on track structures and embankments, causing excessive environmental vibration problems. The excessive vibrations problems have attracted many discussions in theoretical, numerical and experimental aspects [1–5]. For the requirements of high-speed railways of high maintainability, high stability, high reliability, and high running comfort, ballastless track structures have been used gradually in many parts of the world. In 2022, the total high-speed railways reached 30,000 km, covering more than 80% of large cities, of which there are ballastless slab tracks. An analytical method has been investigated considering the couple interaction of track structure and substructures, to study the vibrations of superstructure and substructure [2,3]. The researchers indicated that the dynamic behavior of ballastless slab tracks and subgrades are different from that of ballasted tracks, and need more consideration [6]. The geometrical arrangement of the train and moving speeds influence the dynamic response characteristic of superstructures and substructures (embankment, subsoils, etc.). The mathematical and numerical models of track and ground were presented in both time and frequency domains to study the vibrations on railway track and subsoil, the results of which were verified clearly by field measurement results [7]. A 3D finite element model of

track-subgrade-foundation supported by pile group was developed to study the dynamic stress and deformation distribution in geosynthetic-reinforced pile foundation [8]. The railway weight and moving train loads were simulated as the main vibration exciting vibration sources. Zhang et al. [9] A 3D discrete element model was established, and validated by laboratory tests, to study the ballast deformation behavior under cyclic and high speed train loads. The FEM simulation results shown that the load frequency has impaction on ballast deformation in certain frequency ranges. A 2.5D finite element analysis on vibrations of track and underlying soil foundations was presented using vehicle, track, and foundation coupled model, and found that train speed and track irregularities influence the dynamic response of the ground and track [10,11]. A semi-analytical vehicle-track-ground coupled model was created in homogeneous elastic half-space, and found the impact of train speed, testing locations, rail irregularity, subgrade-bed stiffness, and rail type on ground vibrations [12].

The above studies mainly focus on the vibrations of railway tracks, subgrade, and subsoils in the condition of assuming an interaction between the track and ground. There are few studies on vibrations and long-term durability of the ballastless slab track structure, embankment, and foundation. Field tests and experimental tests can be efficient methods to observe these vibration questions in ballastless high speed railway under different train speeds and loads. A scale model test was developed to study the deformation character on railway and subgrade, two loading mode were selected and applied on rails [13]. Two loading test modes were presented on a scale of 1:5 on the dynamic behavior of ballasted railway tracks, and the cyclic deformations between two loading test methods were compared [14]. A reduced scale (1:3) model of railway track was built to investigate the dynamic settlement and behavior under cyclic train loads [15]. The moving passages of bogies on sleepers at different speeds were simulated and applied using hydraulic jacks, similar to the form of an M wave. A full size steel test box was built to create a physical model of the ballatless railway, subgrade, and subsoil [6]. The dynamic loading process was simulated with a series of exciting actuators by applying loads on tracks to study the dynamic behavior of track structure and subsoil. For ballastless and ballasted tracks, field measurement and test results have a different dynamic response and distribution of stress velocity at different train speeds.

The high development of high-speed railway lines cross through areas with poor geological conditions, such as soft soil, offshore and low-lying marsh areas, resulting geotechnical problems, such as large settlements and reduction of bearing capacity. The running comfort, safety, economy, and low cost of ballastless railway lines require innovative construction and ground improvement techniques. Some researches were presented on physical properties of granite residual soil reinforced by fibers, organic, and inorganic modifiers [16,17]. As a ground improvement technique, pile-soil composite foundations can improve bearing capacity, settlement under static or dynamic loads and have good performance on reinforced soils in terms of stiffness and mechanical properties [18,19]. In recent years, as a new soil reinforcement method in high-speed railway lines, the piled raft structure has been used to improve the soil conditions and control excess settlement. Researchers have done works on the static bearing capacity, pile-soil interaction, and parameter analysis of piled raft structures [20–23], but fewer studies have focused on dynamic behaviors. A shaking table test on piled raft foundations in sand was presented using a geotechnical centrifuge [24]. The testing results shown that the inclination of the piled raft in the shaking process is much smaller than that of the pile group due to the contribution of soil resistance just beneath the raft. A practical method was developed to investigate the dynamic performance of a piled raft foundation [25]. The dynamic contact character and forces on the pile–soil interface were defined using the analytical method to clearly investigate the dynamic response of the piled raft foundation in layered soil. A simplified analytical and FEM method of forming a piled raft foundation model were presented, and the dynamic interaction factor between the pile and raft was established and calculated [26]. A 3D coupled FEM-BEM method was presented to study the dynamic response of piled

raft structure under the vertical and horizontal loads [27]. The dynamic interaction factors between pile group, raft, and soil were obtained and used for the calculation of the piled raft foundation models. A 3D FEM model of CRTS track, subgrade, and geosynthetic-reinforced pile foundation was developed in high speed railway, and the train induced loads were simulated through ABAQUS subroutine VDLOAD and applied in quasi-static load mode [8]. The distribution of dynamic soil stress and deformation in the track and subgrade system were calculated and presented. A large scale model of a ballastless track and piled raft supported system was built in a soil layer of sand with dried and saturated conditions [28]. The parameters such as train speed, soil condition, and loading frequency were used in the loading test to investigate their influences on the characteristics of dynamic responses and resonant frequency of the piled raft supported foundation.

The above researches have done many works in static and dynamic about the railway track, subgrade, and pile supported foundations. Most of them focused on the static and dynamic bearing behavior subjected to static or cyclic loads. Few studies have been carried out to establish the dynamic behavior of piled raft supported ballastless track system in soft soil with different underground water levels.

A scale (1:5) model of the ballastless slab track, embankment, and piled raft foundation was built in the conditions of two underground water levels. The low and high water levels are 1 m and 4.3 m below the subsoil surface, respectively. A dynamic exciter loading system was developed to exert dynamic loads with a shape of M waves on the rails, to simulate the train moving on the rail with a constant speed. Vibration velocities at specified locations for the railway structure and piled raft foundation were obtained and analyzed in time and frequency domains under different water levels.

## 2. Experiment Overview

### 2.1. Experimental Model

A large-scale experimental model (1:5) was conducted using a reinforced concrete tank with the length of 5 m, width of 4 m, height of 7 m, to satisfy the physical and material relationships. The geometric structure layout of the track structure and piled raft foundation is shown in Figure 1. The constructed physical model concludes track structure, rails, roadbed, embankment, pile-raft structure, subsoil, and sand. Two underground water levels for the pile-raft foundation in silty soil are considered. For model cases 1,2, the water levels 1 and 2 were 1 m and 4.3 m below the subsoil surface. The supporting layer with sand is 1 m thickness and arranged at the bottom of the concrete tank. In order to weaken the vibration propagation boundary effect of dynamic loading on the wall of the model tank, the waterproof geotextile and foam cotton are added. Waterproofing of model tank can also be achieved using waterproof geotextiles. The physical model is reduced to a scale of 1:5. The experimental model scale factors were calculated by Bockingham π theorem and shown in Table 1. The physical model keeps the physical and mechanical properties of the materials.

**Table 1.** Experimental model scale factors.

| Parameters | Scale Factors | Parameters | Scale Factors |
|:---:|:---:|:---:|:---:|
| Load | 1:25 | velocity | 1 |
| stress | 1 | time | 1:5 |
| volume | 1:125 | length | 1:5 |
| frequency | 5 | modulus | 1 |
| density | 1 | | |

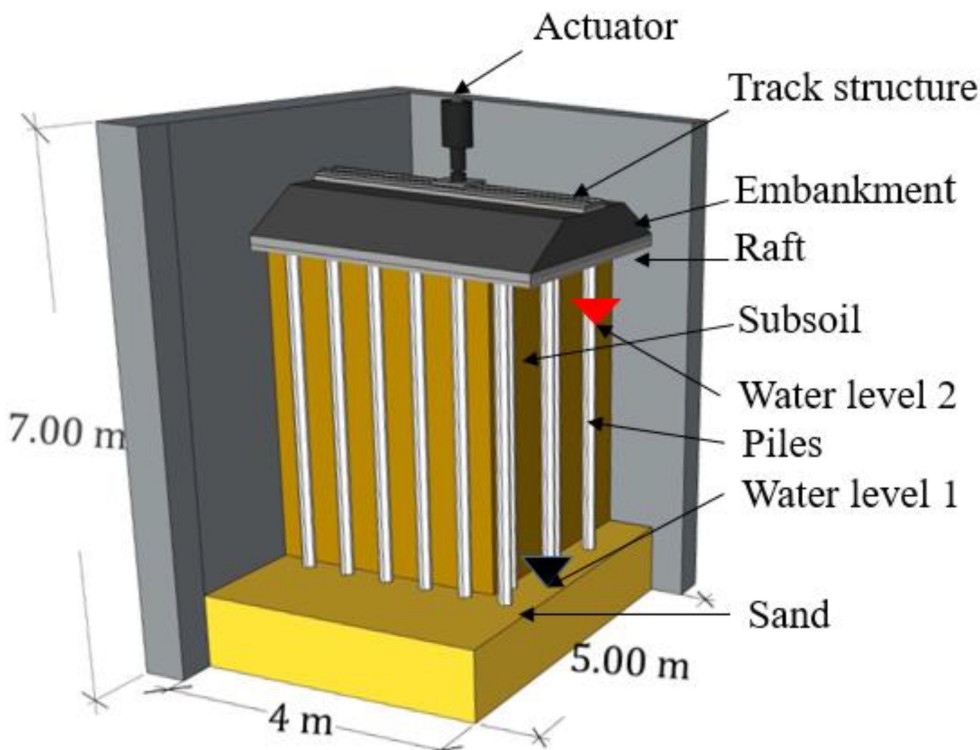

**Figure 1.** Geometric structure layout of track structure and piled raft foundation.

The track slab of the China Railway Track System III (CRTS III) with a dimension of 4.856 m × 2.5 m × 0.19 m thick was designed, prefabricated in a factory and shipped to the experiment site. A concrete base of dimensions 5 × 3.1 m × 0.3 m thick was constructed in situ with steel reinforcement. The roadbed was a layer of 0.08 m thick and filled with gravel to support the concrete base, which was fixed to the track slab. In Figure 2, the embankment is 0.54 m thick and filled with AB Granular. The concrete raft structure and cushion layer have the thickness of 0.12 m, and 0.06 m. Velocity sensors V1, V2, V3, V4, V5, V6 were arranged in the horizontal directions of the piled raft foundation. Velocity sensors V7, V8, V9, V10, V11 were arranged in the vertical directions of the piled raft foundation. V1 to V6 are located on the slab track, roadbed, embankment, and subsoil with a certain distance from the track center. V7, V8, V9, V10, V11 are located on the embankment top, raft top, subsoil top, middle of subsoil, and bottom of the subsoil along the depth in the middle cross section of the testing model.

*2.2. Applied Dynamic Load*

A synchronous excitation and loading system was developed to simulate the dynamic loading test process [6]. The loading curves are calculated from the fasteners reaction force, and changed with time t and speed v for different train moving speed. Cyclic dynamic loads were applied to the track slab through actuators controlled by a hydraulic private service. The testing results of dynamic response shown frequency character mainly between 0 Hz and 30 Hz. The geometry configuration of the standard carriage of a China Railways High-speed train is shown in Figure 3a. Sun et al. [28] created a dynamic loading curve with the shape of M waves to simulate the train moving on the railway in the model test. This paper chose a Fourier equation to express the cyclic load $F(t)$ corresponding to time t and circular frequency $\omega$.

$$F(t) = a_0 + a_1 cos(\omega t) + b_1 sin(\omega t) + a_2 cos(2\omega t) + b_2 sin(2\omega t) + a_3 cos(3\omega t) + b_3 sin(3\omega t) \tag{1}$$

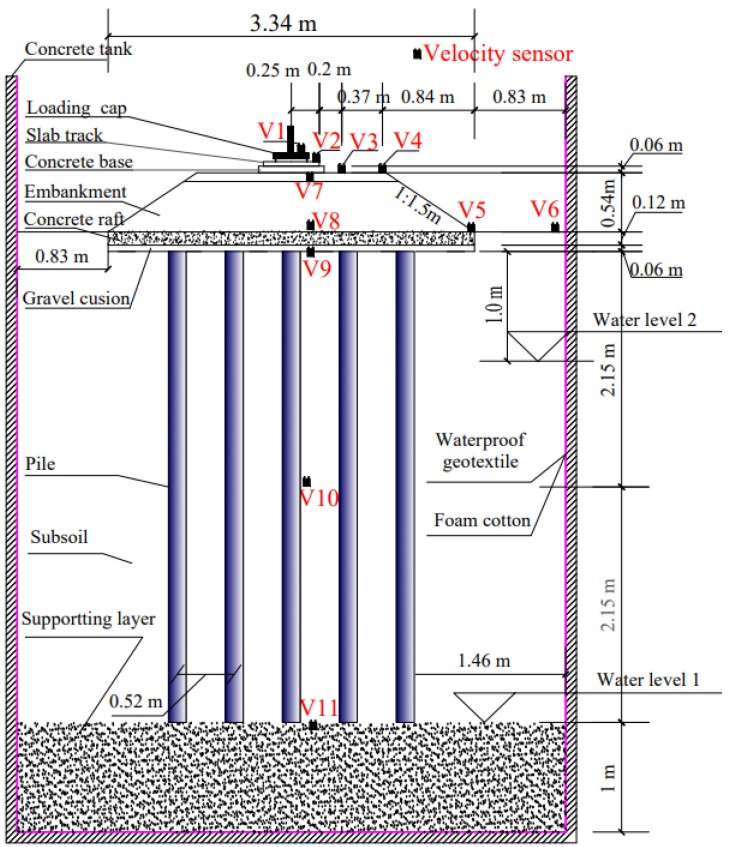

**Figure 2.** Layout of experimental model.

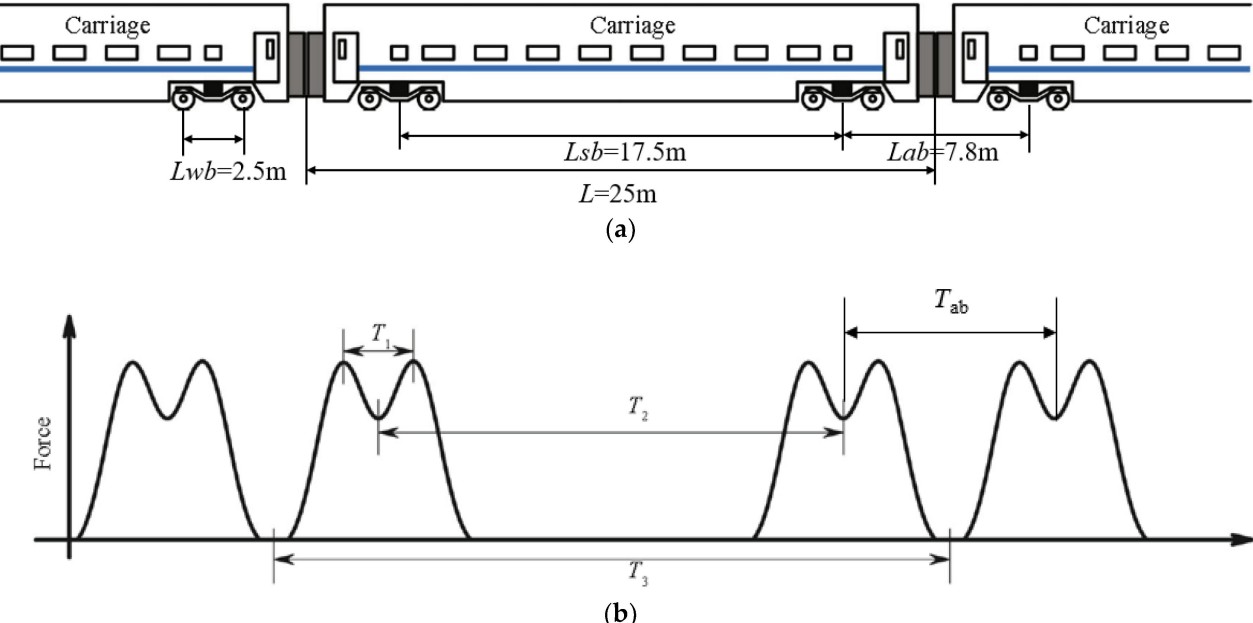

**Figure 3.** Geometry configuration of the high-speed trains and M-shaped wave. (**a**) Geometry configuration of high-speed trains; (**b**) M-shaped wave.

Note: The analytical expression of the force was calculated from the Fourier equation, where the circular frequency $\omega = 2\pi f$ (rad/s); $f = v/Lab$ is the frequency (Hz); v is the train speed (km/h); L is the length of train (L = 25 m in full scale); Lab is the distance between two adjacent bogies; Lwb is distance between two wheels. M-shaped wave

considering the entire carriage effect is shown in Figure 3b. T3 is the cycle time (s); T1 is the time of two wheel loads moving at speed v; f1 = 1/T1 is corresponding to the maximum exciting frequency. $F(t)$ is wheel load. In the analytical analysis process, the corresponding $\omega = 2\pi f = 2\pi v/Lab$, so the circular frequency $\omega$ are 40.25, 60.39 respectively for train speed v = 180 km/h, v = 270 km/h. For $F(t)$ = 160 kN, the fitting coefficient are calculated through three-order Fourier series equation, and shown as: $a_0 = 85.52$, $a_1 = -71.84$, $b_1 = 15.24$, $a_2 = -32.29$, $b_2 = 14.97$, $a_3 = 12.89$, $b_3 = -10.45$. The dynamic load induced by the exciter in the experiment can simulate various frequency contents corresponding to the geometry configuration of the train passage.

The simulated axle load of the harmonic train in full scale is125 kN (12.5 t). So the input load magnitude in the exciter is set to 5 kN following the size scale factor 1:25. Loading curves at reduced scale for train speed of 180 km/h and 270 km/h in time and frequency domain are shown in Figure 4. Dynamic loading tests under the M-shape wave load with frequencies varying from 1–30 Hz were carried out to determine the effect of cyclic frequency on the piled raft foundation.

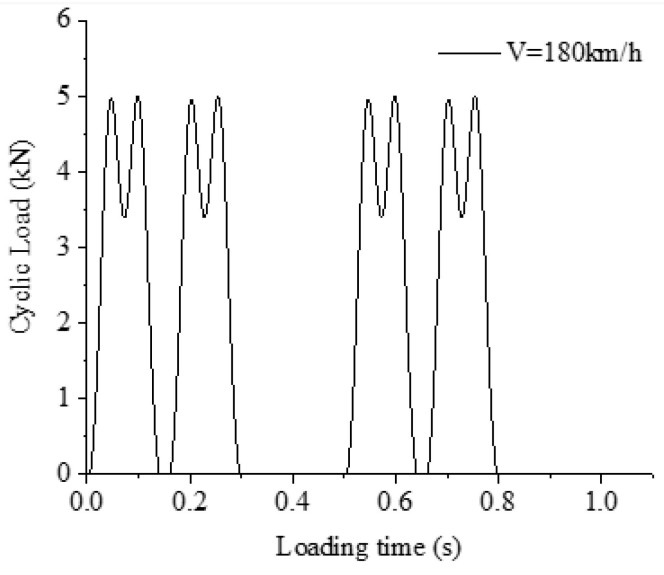

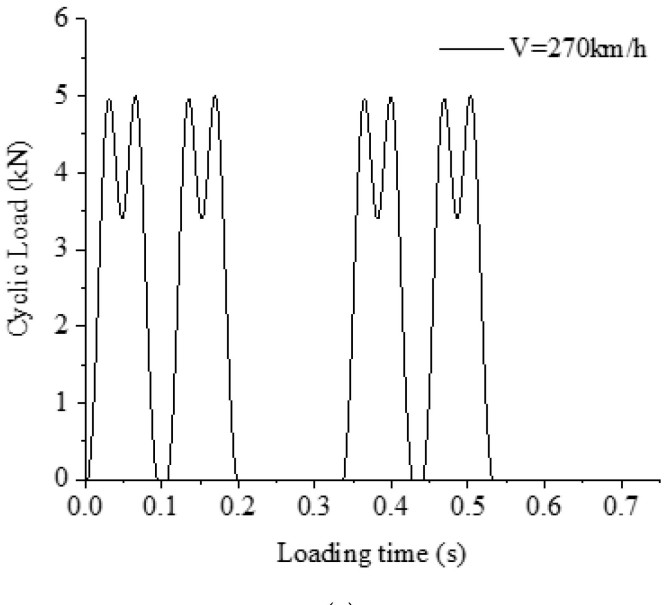

(a)

**Figure 4.** *Cont.*

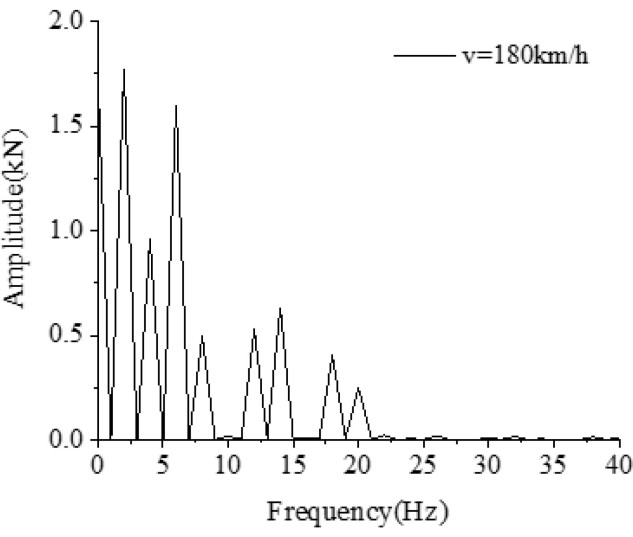

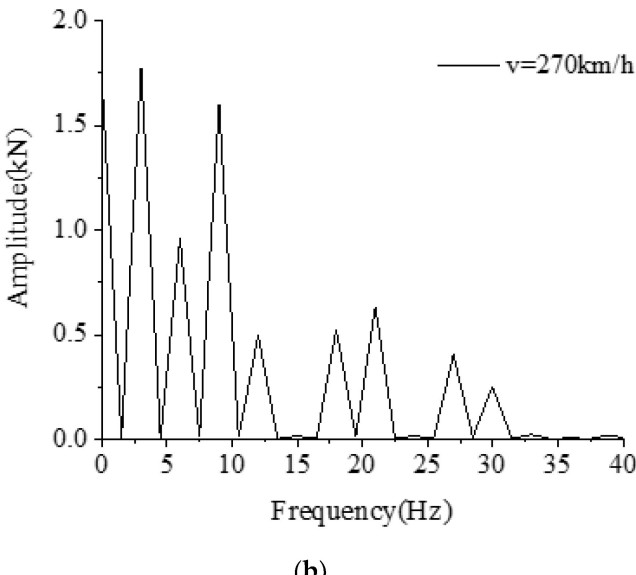

(**b**)

**Figure 4.** Loading curves at reduced scale for train speed of 180 km/h and 270 km/h in time and frequency domain. (**a**) time domain; (**b**) frequency domain.

### 3. Results and Discussions

*3.1. Analysis of Vibration Velocities in Time and Frequency Domain*

Two water levels for the piled raft foundations in silty soil are considered, where the low and high water levels are 5.3 m, 1 m below subsoil surface. Testing results about the vibration velocities at the testing points (shown in Figure 2) are obtained and discussed. The time histories of vibration velocities at V2, V3, and V6 for train speed v = 180 m/h, v = 270 km/h are plotted in Figures 5 and 6. In Figure 5a, for V2, V3, the time history and peak velocity induced by train load are clear and have local drastic fluctuations. For the track slab and roadbed, the peak value of the vibration velocity caused by train wheel load is clearly visible. The time histories of vibration velocity at the track slab and roadbed have the variation pattern similar to "M shaped" waves, which correspond to the dynamic loading curves. In Ref [28], the variation pattern of vibration velocity corresponds to loading on the bogies, the time history curves have the similar shape of letter "M". The maximum velocity at the track slab is much stronger than that at roadbed, and decreases dramatically by about 88%, from 25.9 mm/s to 2.9 mm/s. In Figure 5b, for the high water

level case, the vibration velocities at the above locations (V2, V3, V6) follow a pattern similar to that for the low water case, but have differences in peak values.

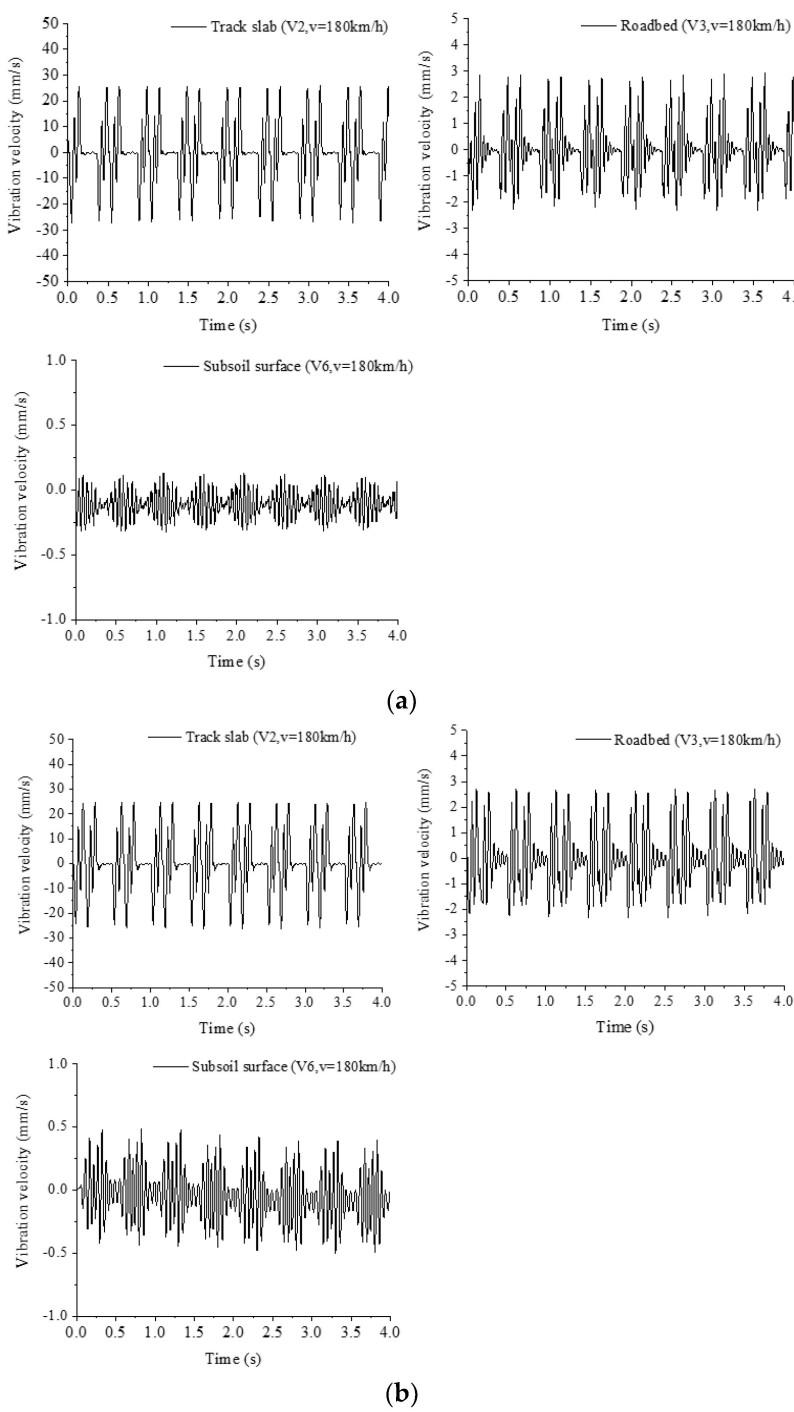

**Figure 5.** Time histories of vibration velocity at track slab V2, roadbed V3, and subsoil surface V6. (**a**) Low water level v = 180 km/h, (**b**) High water level, v = 180 km/h.

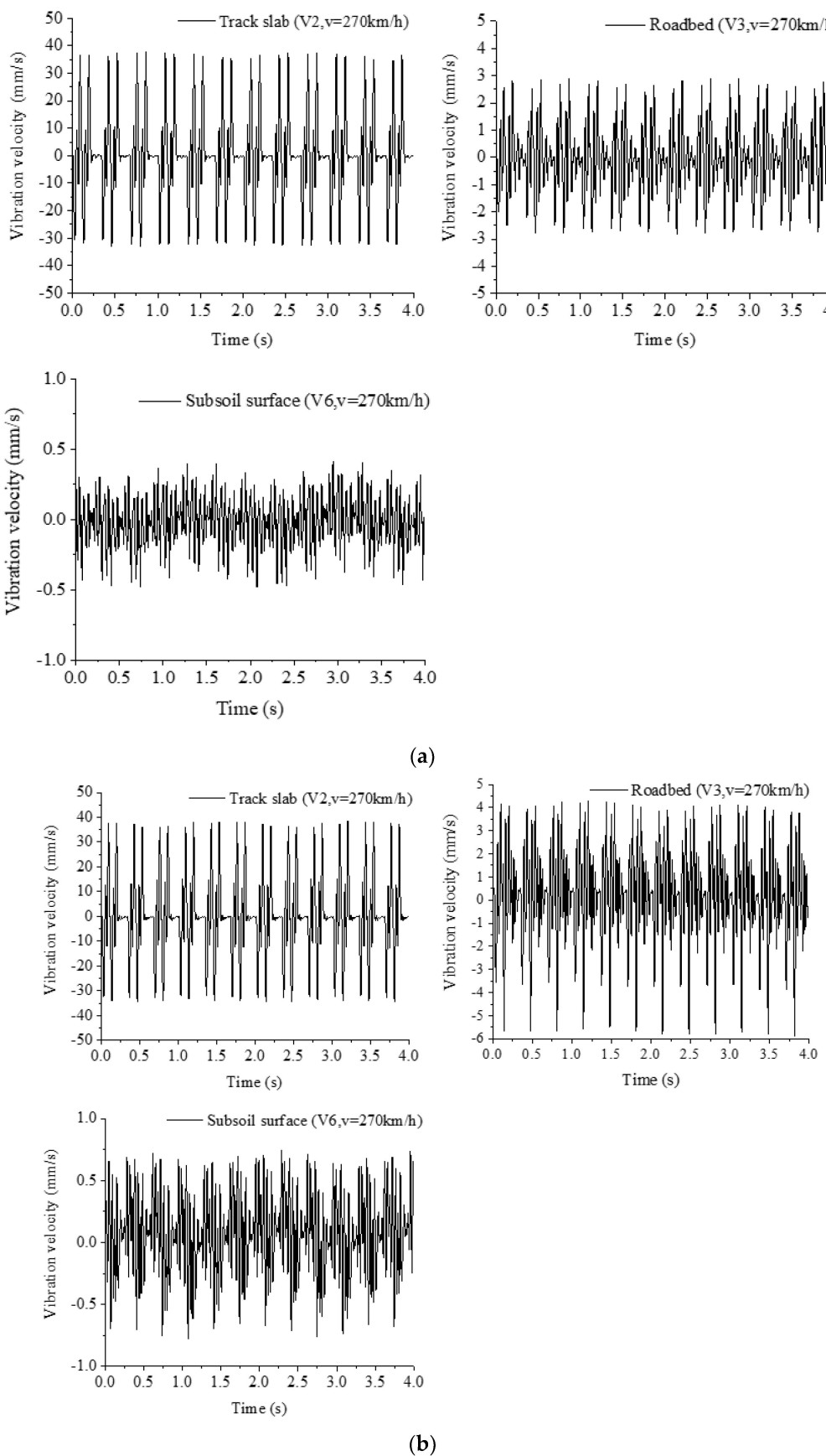

**Figure 6.** Time histories of velocity at track slab V2, roadbed V3, and subsoil surface V6. (**a**) Low water level, v = 270 km/h, (**b**) High water level, v = 270 km/h.

Figure 6 shows the time histories of vibration velocities V2, V3, V6 for v = 270 km/h under low and high water level conditions. Compared to Figure 5, the vibration velocities at V2, V3, and V6 increase with the train speed and have a relatively large increase at the track slab. More sharp impulses and peak points in of time history curves occur with the increasing train speed. The water level change may have little impaction on the surface vibration velocity of track structure, embankment, and subsoil. The increase of vibration velocity decreases with the increase of the distance away from the track slab.

Figures 7 and 8 show the frequency contents of velocity at V2, V3, V6 for loading speed v = 180 m/h, v = 270 km/h. The dynamic frequency response contents show the main features of the railway structure and piled raft supported foundation in the ranges of 0–35 Hz for two water level conditions. In Figure 6, the frequency spectral response curves show peaks at 2 Hz, 4 Hz, 6 Hz, 8 Hz, 12 Hz, 18 Hz, 20 Hz for train speed v = 180 km/h, and the frequencies 2 Hz, 6 Hz, and 20 Hz correspond to one carriage length of 25 m, a distance of 7.5 m between adjacent bogies, and a distance of 2.5 m between two wheels. These frequencies such as 4 Hz, 6 Hz, 8 Hz, 12 Hz, and 18 Hz are the dominant harmonic frequencies and show peak frequency spectrum characteristics of V2, V3. However, the dominant harmonic frequencies of vibration velocity at the subsoil surface are mainly concentrated at 18 Hz and 20 Hz. The dynamic energies at frequency below 18 Hz are dissipated and absorbed by the upper embankment. The superstructure (track slab, roadbed) and substructure (pied raft structure) can attenuate and absorb the dynamic power induced by the upper train vibration load. As mentioned in Figure 5, in the condition of high water level, the water level has slight influence on the vibration velocity. The similar result can also be found in Figure 7b. Most of the peak frequency spectral for V2, V3 are concentrated at frequencies of 2 Hz, 4 Hz, 6 Hz, 8 Hz, 12 Hz, 18 Hz, and 20 Hz. The increase of water level has a certain influence on the frequency distribution and peak value of vibration velocity at the subsoil surface. Studies on the relationship between vibration frequency response characteristics and train structure arrangement have also given similar results from field measurement [29] and model tests [6,30].

Similar phenomenon about the frequency spectral characteristic can also be found in Figure 8 for train speed v = 270 km/h, and there are other differences. In Figure 8, the dynamic frequency response contents show peaks at 3 Hz, 6 Hz, 9 Hz, 12 Hz, 18 Hz, 21 Hz, 27 Hz, 30 Hz, of which 3 Hz, 9 Hz, and 30 Hz correspond to one carriage length of 25 m, the adjacent bogies distance of 7.5 m and wheel distance of 2.5 m. Most of the frequency spectrum of vibration velocities distribute in frequencies below 30 Hz. These peaks such as 6 Hz, 9 Hz, 12 Hz, 18 Hz, 21 Hz, 27 Hz are the dominant harmonic frequencies and show peak frequency spectrum characteristics of the track structure and embankment. The dominant harmonic frequencies of vibration velocity at the subsoil surface are mainly concentrated at 18 Hz and 21 Hz in the condition of low water level. The increase of water level changes the peak values and frequency spectrums at the corresponding frequencies in the range of 0 Hz to 30 Hz. In Figure 8b, the dominant harmonic frequencies of velocity response at the subsoil surface are mainly concentrated at 3 Hz, 18 Hz, 21 Hz, 27 Hz, and 30 Hz in the condition of high water level. From this, we can know that the time and frequency character of vibration velocity at specified locations are determined by the geometry, load, and speed of the train.

In Figures 9 and 10, the vibration velocities at embankment top V7, raft top V8, and subsoil top V9 in the middle cross section of the model were presented. For train speed v = 180 km/h, the vibration velocity levels inside the embankment and subsoil are lower than that on the surface of the track structure and embankment, but still have visible impulse. The vibration velocity decreases when it crosses the embankment, reducing by about 42% and 48%, respectively, for low and high water level conditions, to a value below 1 mm/s at the location of subsoil top. Water level change has little impact on the vibration peaks of piled raft supported foundation in the time domain.

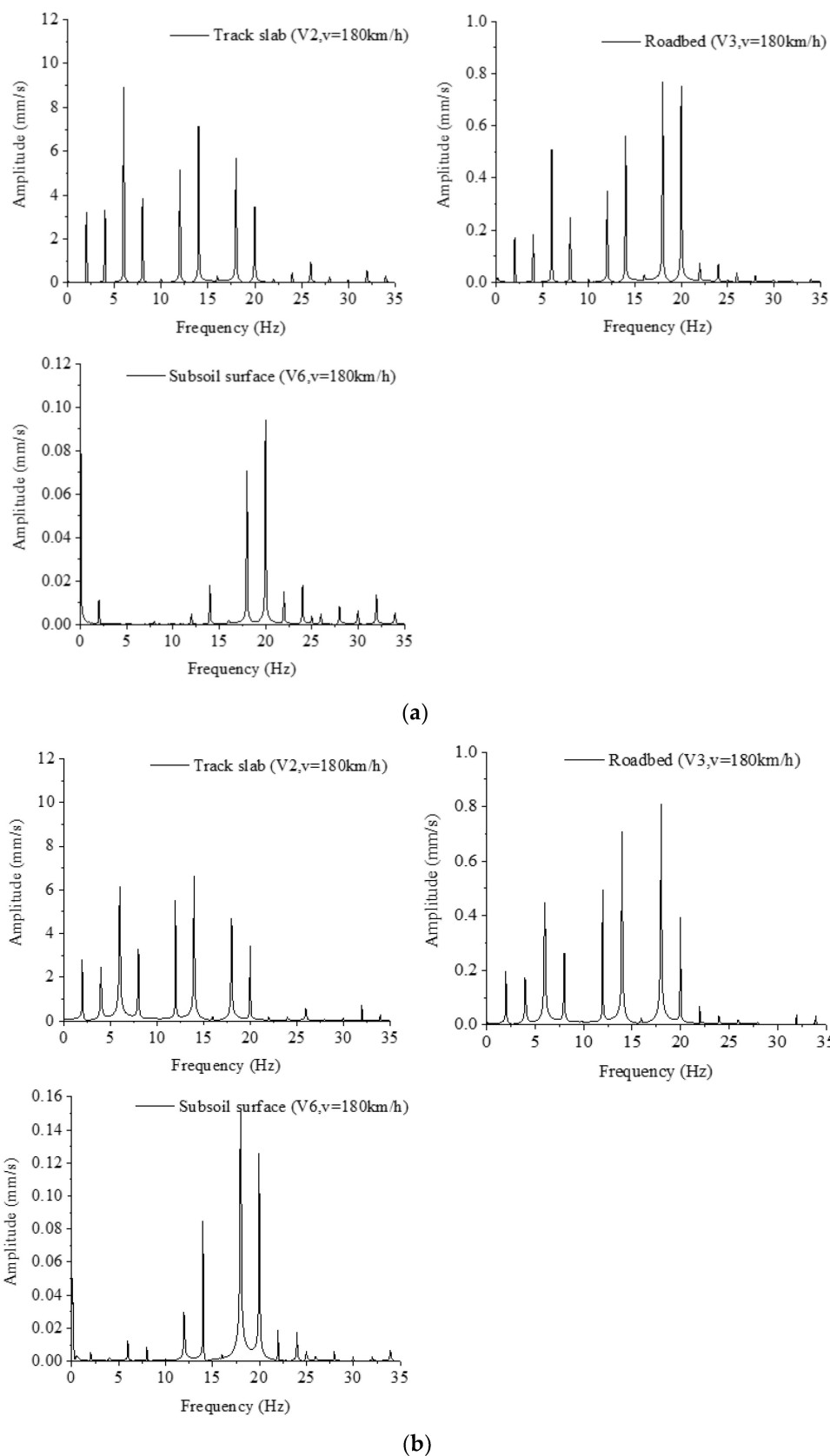

**Figure 7.** Frequency contents of velocity at track V2, V3, V6. (**a**) Low water level, v = 180 km/h, (**b**) High water level, v = 180 km/h.

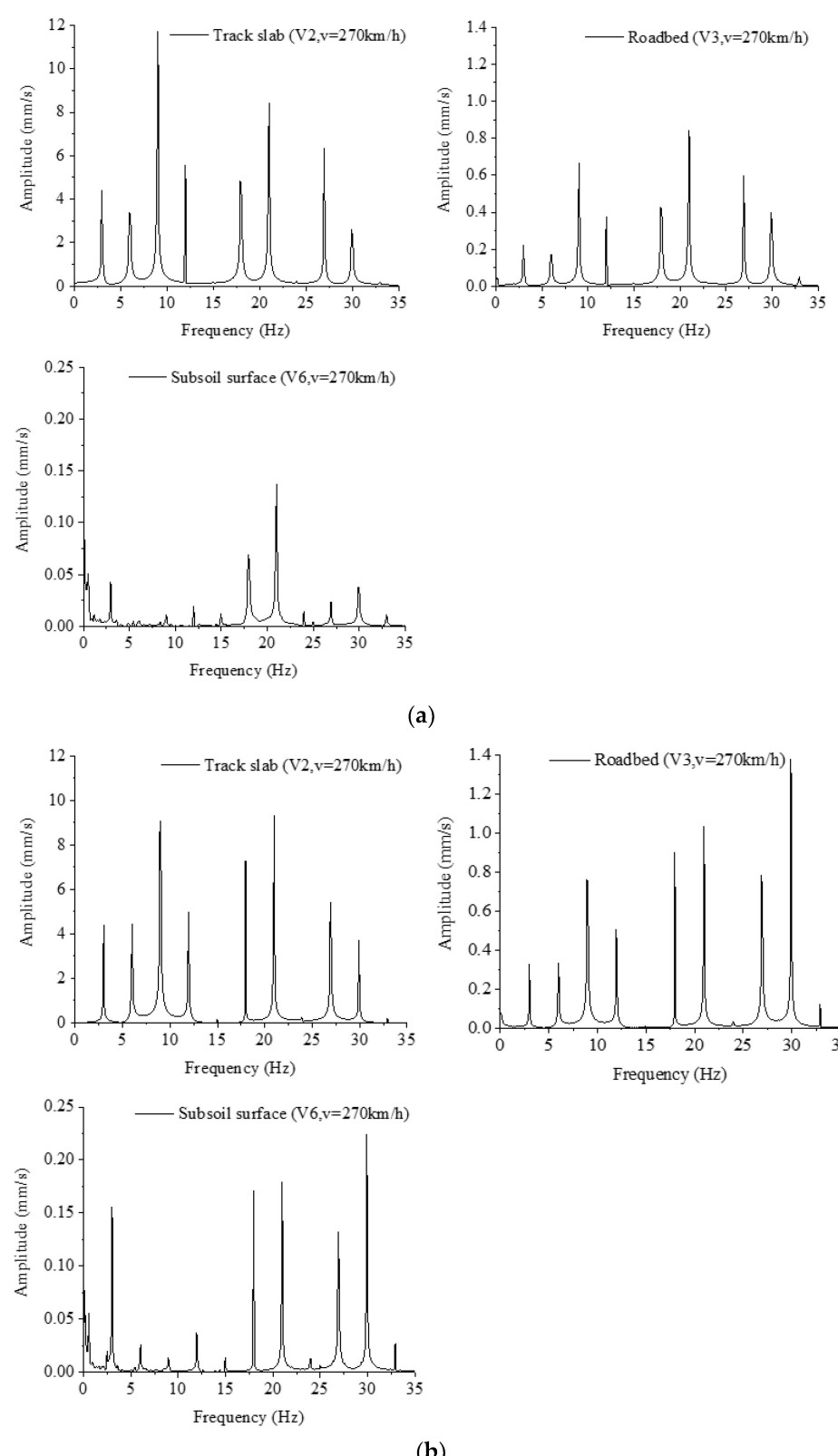

**Figure 8.** Frequency contents of velocity at track V2, V3, V6. (**a**) Low water level, v = 270 km/h, (**b**) High water level, v = 270 km/h.

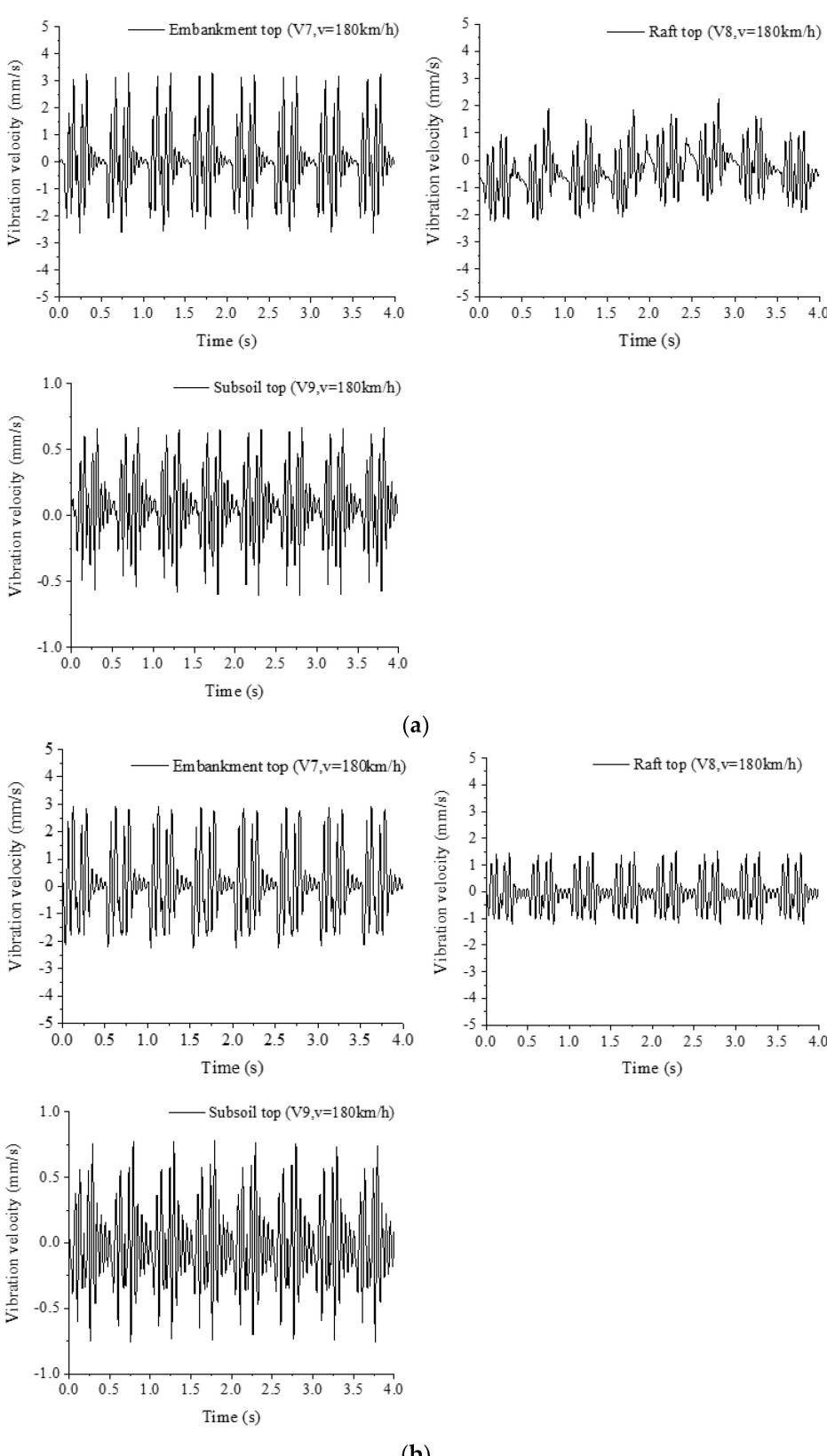

**Figure 9.** Time histories of vibration velocities at embankment top V7, raft top V8, and subsoil top V9. (**a**) Low water level v = 180 km/h, (**b**) High water level, v = 180 km/h.

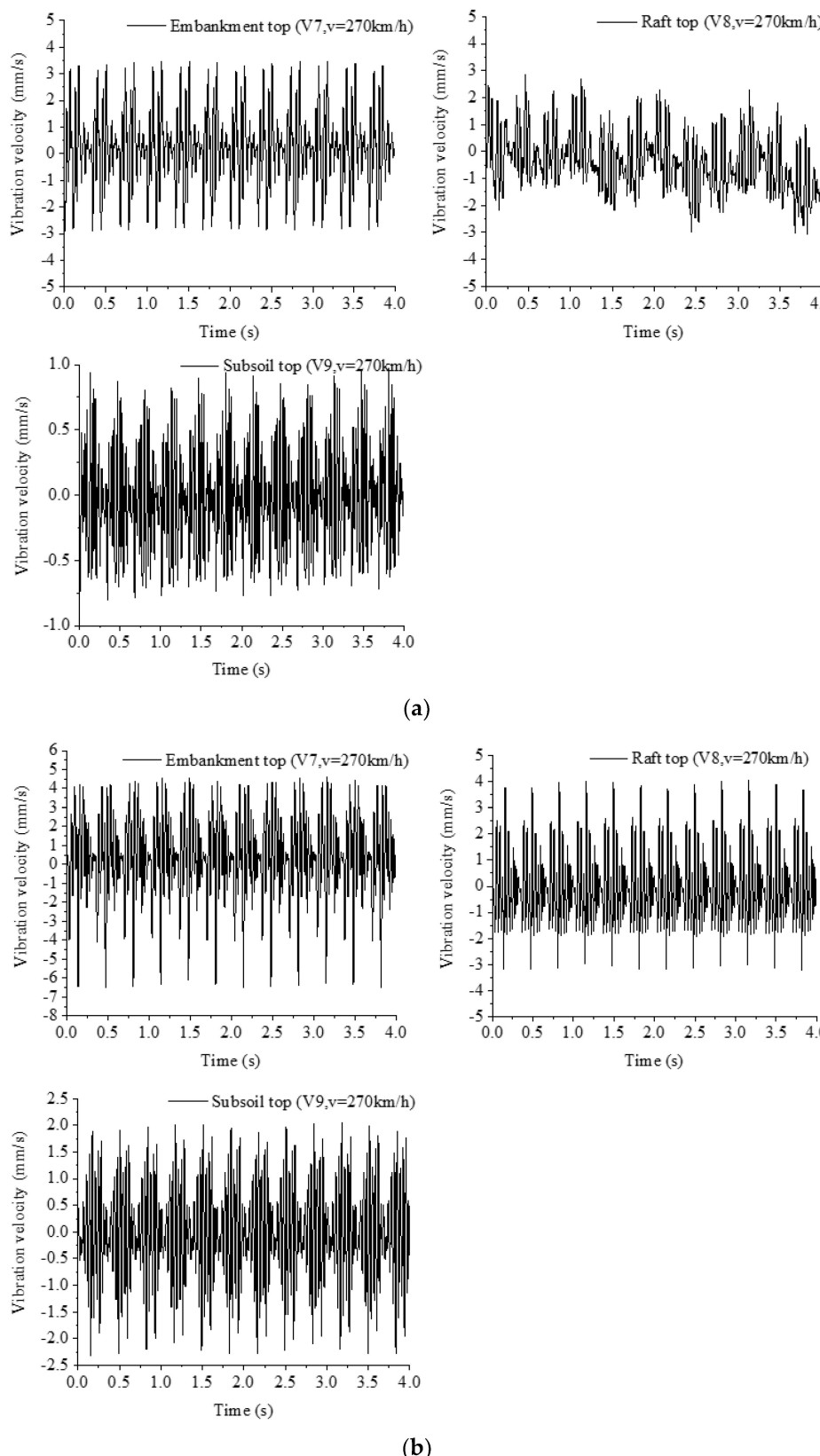

**Figure 10.** Time histories of vibration velocities at embankment top V7, raft top V8, and subsoil top V9. (**a**) Low water level v = 270 km/h, (**b**) High water level, v = 270 km/h.

In Figure 10, v = 270 km/h, the vibration velocity levels inside the embankment and subsoil are lower than that on the surface of the track structure and embankment, but still have visible impulse. With the train speed increase, the peak values at V7, V8, V9 increase accordingly, and have a sharp impulse stronger than that in Figure 9, especially at subsoil top V9.

In Figure 11, the peak frequency spectrum points are distributed in the frequency region below 20 Hz. For v = 180 km/h, the frequency contents of vibration velocities at various locations are distributed at frequency points 2 Hz, 4 Hz, 6 Hz, 12 Hz, 14 Hz, 18 Hz, 20 Hz, corresponding to the harmonic frequencies. The change of water level has impaction on the peak spectrum of vibration velocity at harmonic frequencies, but not obviously. The same results can also be found in Figure 12. Since the frequency distribution ranges from 0 Hz to 30 Hz, the high frequency contents were eliminated by the low-pass digital filter. Vibration velocity frequency contents can reflect the frequency characteristic of loading waves both in frequency range and amplitude. The vibration absorption and attenuation of the embankment, raft, and subsoil also influence the vibration load transmission and attenuation. Therefore, the dynamic frequency response character of the track slab, embankment, piled raft, and subsoil are dominated by the dimensions of trains, properties of vibration medias, and load excitation sources.

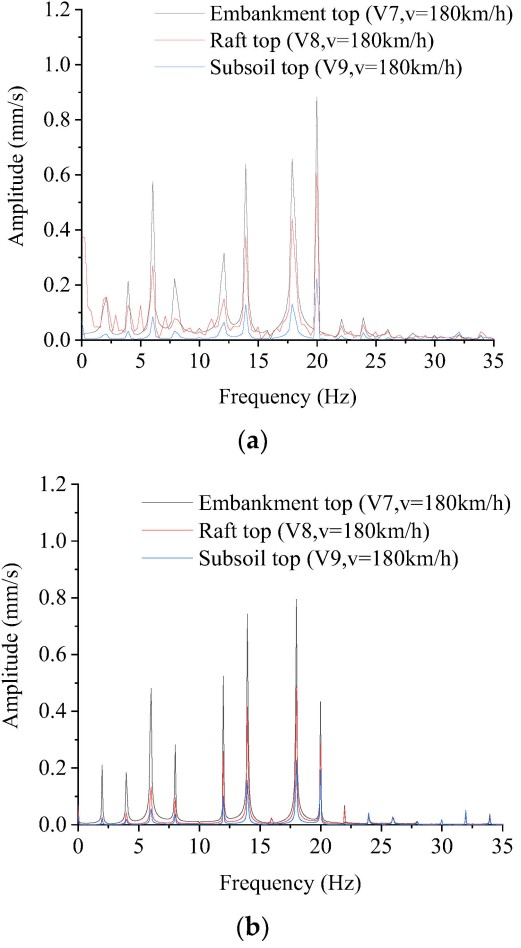

**Figure 11.** Frequency contents of vibration velocities at embankment top V7, raft top V8, and subsoil top V9. (**a**) Low water level, (**b**) High water level, v = 180 km/h.

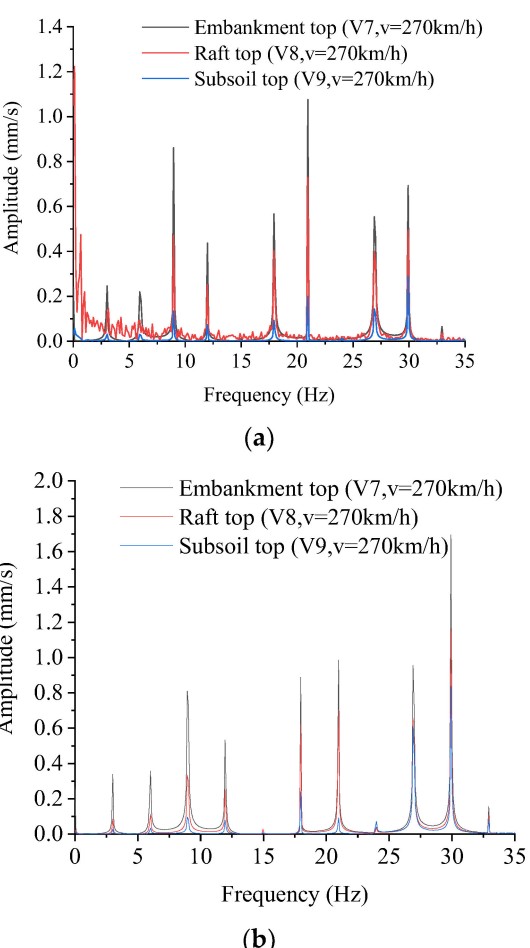

**Figure 12.** Frequency contents of vibration velocities at embankment top V7, raft top V8, and subsoil top V9. (**a**) Low water level, (**b**) High water level, v = 270 km/h.

*3.2. Distribution of Vibration Velocities in Piled Raft Foundation*

Figure 13 shows the peak velocity of V2, V3, V4, V5, V6 in the horizontal direction from the track center for three train speeds in low and high water level cases in silty soil. In Figure 13a, the low water level is 1 m below the base of the silty subsoil, the peak vibration velocities on the track slab are 10.3 mm/s, 25.9 mm/s, 37.8 mm/s for v = 90 km/h, v = 180 km/h, v = 270 km/h, and reduce by 88.8%, 88.6%, 92.3% at the roadbed, respectively. The peak vibration velocities on roadbed surface are 1.2 mm/s, 2.9 mm/s, and 2.9 mm/s for v = 90 km/h, v = 180 km/h, v = 270 km/h. Filed and model test results of peak velocity on roadbed for ballastless high speed railway are around 1.9 to 9.64 mm/s [6,31]. Vibration peak velocities on the track slab are larger than other locations and have drastic reduction on roadbed. The attenuation laws of the vibration velocities for different train speeds are consistent with each other. For v = 90 km/h, v = 180 km/h, v = 270 km/h, the peak velocities on the subsoil surface are 0.08 mm/s, 0.15 mm/s, 0.5 mm/s, and decrease by 93.2%, 94.9%, 82.1% compared with that on roadbed. The peak velocities on subsoil surface are lower than those on the roadbed top, roadbed shoulder, and slope toe. A full-scale model testing results of the vibration velocities at the track structure and roadbed were compared with the field measurements, indicated that the model testing results have good agreement with the field measurements [6]. The vibrations are found strongest at the track slab and reduced by 60.6% and 67.5% at the roadbed at train speeds of 216 km/h, 108 km/h. In this paper, compared with reference [6], the vibration velocity has a large attenuation rate and amplitude. One of the possible influencing factors may be whether the foundation is reinforced or not by the piled raft structure.

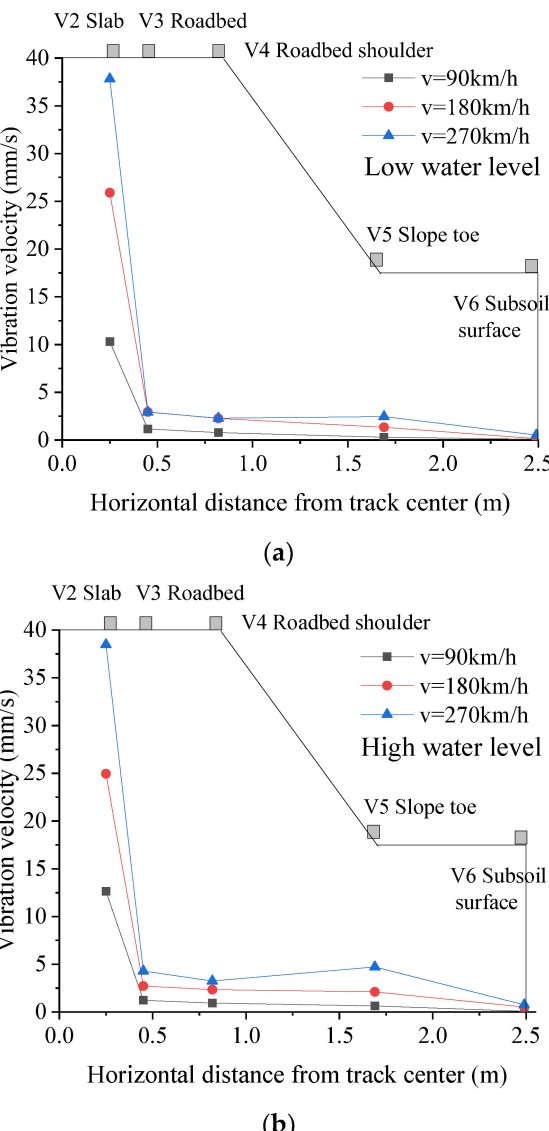

**Figure 13.** Vibration velocity distribution in horizontal direction from the track center. (**a**) low water level, (**b**) high water level.

In Figure 13b, the high water level is 1 m below the surface of silty subsoil, the peak velocities on the track slab are 12.6 mm/s, 24.9 mm/s, 38.5 mm/s for v = 90 km/h, v = 180 km/h, v = 270 km/h, and decrease by 90.3%, 89.1%, 88.9% on roadbed top respectively. The peak velocities on the roadbed top are 1.2 mm/s, 2.7 mm/s, 4.3 mm/s, and are lower than that on the track slab. Similar attenuation law of vibration velocities at other positions for different train speed can be found. With the increase of water level in the subsoil, the peak velocities on the surface of the track slab, roadbed, embankment, and subsoil increase to a certain extent for different train speeds, but not obvious. The peak velocities of track structure are not sensitive to the increase of water level.

Vibration velocity distribution in the horizontal direction for three train speeds can be fitted with an exponential equation,

$$y = A1exp(-x/t1) + y0 \tag{2}$$

The fitted parameters and curves are shown in Table 2 and Figure 14 respectively, where y are the vibration velocity (unit, mm/s); x is the distance away from the track center (unit, mm). For condition of low water level (LWL), the vibration velocity attenuations clearly follow the exponential curve distribution at different train speeds. The peak velocity

y is directly related to the distance x, decreases much quickly with distance up to 0.45 m and slows down from 0.45 m to 1.66 m, with a very low value at location x = 2.49 m. The dynamic loading effect on subsoil surface away from the track slab is lower than the substructures below the track structure. For the condition of high water level (HWL), the vibration velocity attenuations also follow the distribution law of exponential curve at different train speed, but have differences at locations V3, V5. The measured peak velocities at location V3 are lower than the fitting results. The attenuation ranges of vibration velocity from track slab to roadbed are larger than others. The measured peak vibration velocities at location V5 are larger than the fitting results, which indicate that the vibration velocity of the slope has excitation increase phenomenon. The dynamic response character of the embankments is affected by their self-vibration characteristics and dynamic bearing capacity of piled raft structure. The piled raft structure produce a resistance and excitation effect on the vibration of upper embankment materials.

**Table 2.** Fitting parameters of vibration velocities in horizontal direction.

| Model | | ExpDec1 | | | | |
|---|---|---|---|---|---|---|
| Equation | | $y = A1exp(-x/t1) + y0$ | | | | |
| Train speed | v = 90 km/h | | v = 180 km/h | | v = 270 km/h | |
| Water level | LWL | HWL | LWL | HWL | LWL | HWL |
| y0 | 0.38 | 0.078 | 1.24 | 0.525 | 1.75 | 0.744 |
| A1 | 238 | 51 | 687 | 128 | 2619 | 134 |
| t1 | 0.079 | 0.178 | 0.075 | 0.151 | 0.058 | 0.196 |
| Reduced Chi-Sqr | 0.129 | 4.55 | 1.113 | 11.23 | 1.141 | 58.6 |
| RSquare (COD) | 0.9966 | 0.92 | 0.9953 | 0.947 | 0.9978 | 0.883 |
| Adjusted RSquare | 0.9933 | 0.841 | 0.9906 | 0.895 | 0.9957 | 0.766 |

Peak velocity decrease percent from track structure to roadbed, embankment, piled raft, and subsoils in horizontal direction along the track center are calculated and shown in Table 3. For train speed v = 90 km/h, v = 180 km/h, v = 270 km/h, the peak velocities decrease by 88.83%, 88.65%, and 92.28%, respectively from track slab (V2) to roadbed (V3). The decrease percent from track slab to roadbed is larger and close to 90% for different train moving speeds. Vibration energy attenuations mainly occur when crossing the track and roadbed, and slow down, which have little change from V5 to V6. The velocity attenuation characteristics of track slab, roadbed, embankment, and subsoils are different due to the material properties of them in the dynamic loading test. Similar results and laws can also be found in the condition of HWL.

**Table 3.** Peak velocity decrease percent in horizontal direction.

| Train Speed | v = 90 km/h | | v = 180 km/h | | v = 270 km/h | |
|---|---|---|---|---|---|---|
| Water Level | LWL | HWL | LWL | HWL | LWL | HWL |
| Locations | Percent | Percent | Percent | Percent | Percent | Percent |
| V2 | 0.00% | 0.00% | 0.00% | 0.00% | 0.00% | 0.00% |
| V3 | 88.83% | 90.30% | 88.65% | 89.13% | 92.28% | 88.86% |
| V4 | 92.37% | 92.54% | 91.24% | 90.61% | 93.97% | 91.55% |
| V5 | 97.12% | 95.00% | 94.85% | 91.57% | 93.52% | 87.72% |
| V6 | 99.24% | 99.38% | 99.42% | 97.89% | 98.62% | 98.06% |

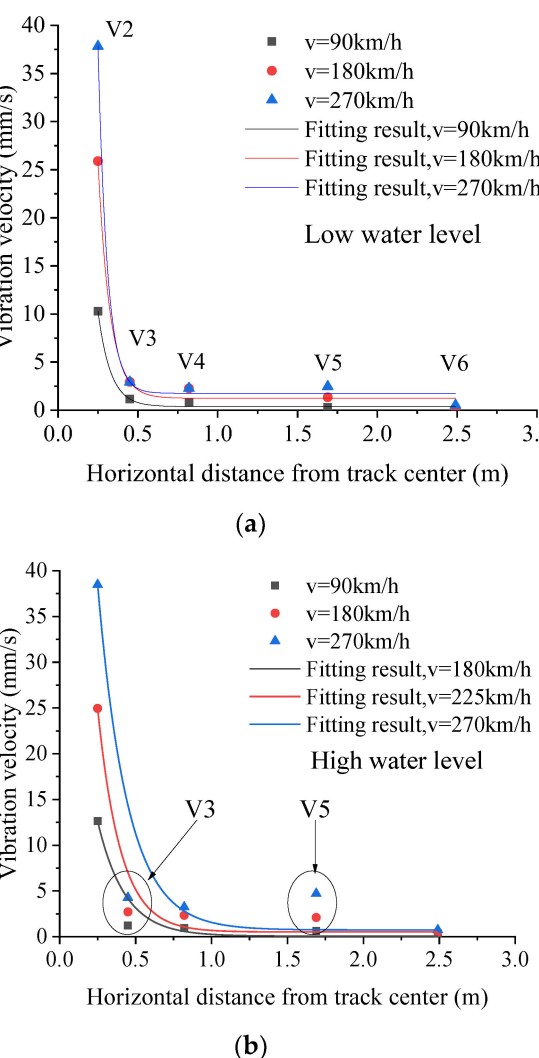

**Figure 14.** Fitting curves of peak velocity in horizontal direction along with distance from the track center. (**a**) Low water level, (**b**) High water level.

Figure 15 and Table 4 show the distribution and decrease percent of peak velocity along the depth of roadbed surface.

**Table 4.** Peak velocity decrease percent in vertical direction along the depth.

| Train Speed | v = 90 km/h | | v = 180 km/h | | v = 270 km/h | |
|---|---|---|---|---|---|---|
| Water Level | LWL | HWL | LWL | HWL | LWL | HWL |
| Locations | Percent | Percent | Percent | Percent | Percent | Percent |
| V7 | 0.00% | 0.00% | 0.00% | 0.00% | 0.00% | 0.00% |
| V8 | 63.01% | 66.53% | 42.25% | 48.06% | −38.11% | 11.85% |
| V9 | 80.08% | 84.38% | 80.33% | 73.42% | 70.86% | 55.40% |
| V10 | 91.75% | 92.25% | 90.28% | 88.34% | 73.25% | 90.68% |
| V11 | 96.75% | 94.73% | 91.06% | 92.48% | 76.02% | 92.65% |

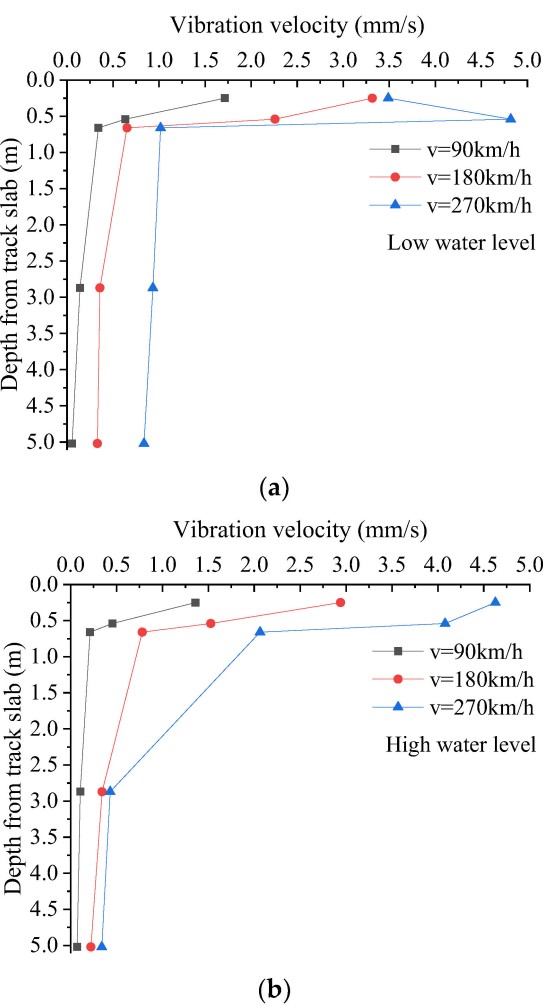

**Figure 15.** Vibration velocity distribution in vertical direction along the depth from roadbed. (**a**) low water level, (**b**) high water level.

For low and high water level conditions, the dynamic velocities decrease quickly within the first 0.54 m away from the roadbed, and then come down. For HWL case, the decrease percent of velocity at subsoil top are 84.4%, 74.4%, 55.4% at train speed 270 km/h, and decrease with the increase of train speeds. The train speeds have more impaction on the vibration attenuation in both track structures and substructures (roadbed, embankment, piled raft structure, and subsoil). In Figure 15a, with the increase of soil depth, the magnitude difference of vibration velocity in the subsoil is getting smaller along the soil depth and has low values. In Figure 15b, for train speed v = 270 km/h, the vibration velocity on the subsoil surface is 2.06 mm/s, which has a certain increase compared to the low water level condition, while the increase of ground water level has little effect on the amplitude changes of vibration velocity at other locations. The piled raft structure has a vibration barrier effect on the loading transfer from embankment to subsoil foundation.

*3.3. Influence of Train Speeds*

In Figure 16, the peak velocities on the track slab increase linearly with the increase of train speed, which are larger than other places such as roadbed, slope toe, and subsoil surface. Train speed changes have more clear impact on velocity response of track slab. In Figure 16a, for V2,V3, V4, V5, V6, the peak velocities increase along with the train speed from 45 km/h to 225 km/h, and decrease slowly except for V5, V6. In Figure 16b, for V2, V3, V4, V5, V6, the peak velocity increase with the train speed from 45 km/h to 270 km/h. For low and high water level cases, the velocities at these locations show a similar ascendant

tendency except for V3, V4. The vibration interference superposition effect is a reasonable explanation about this.

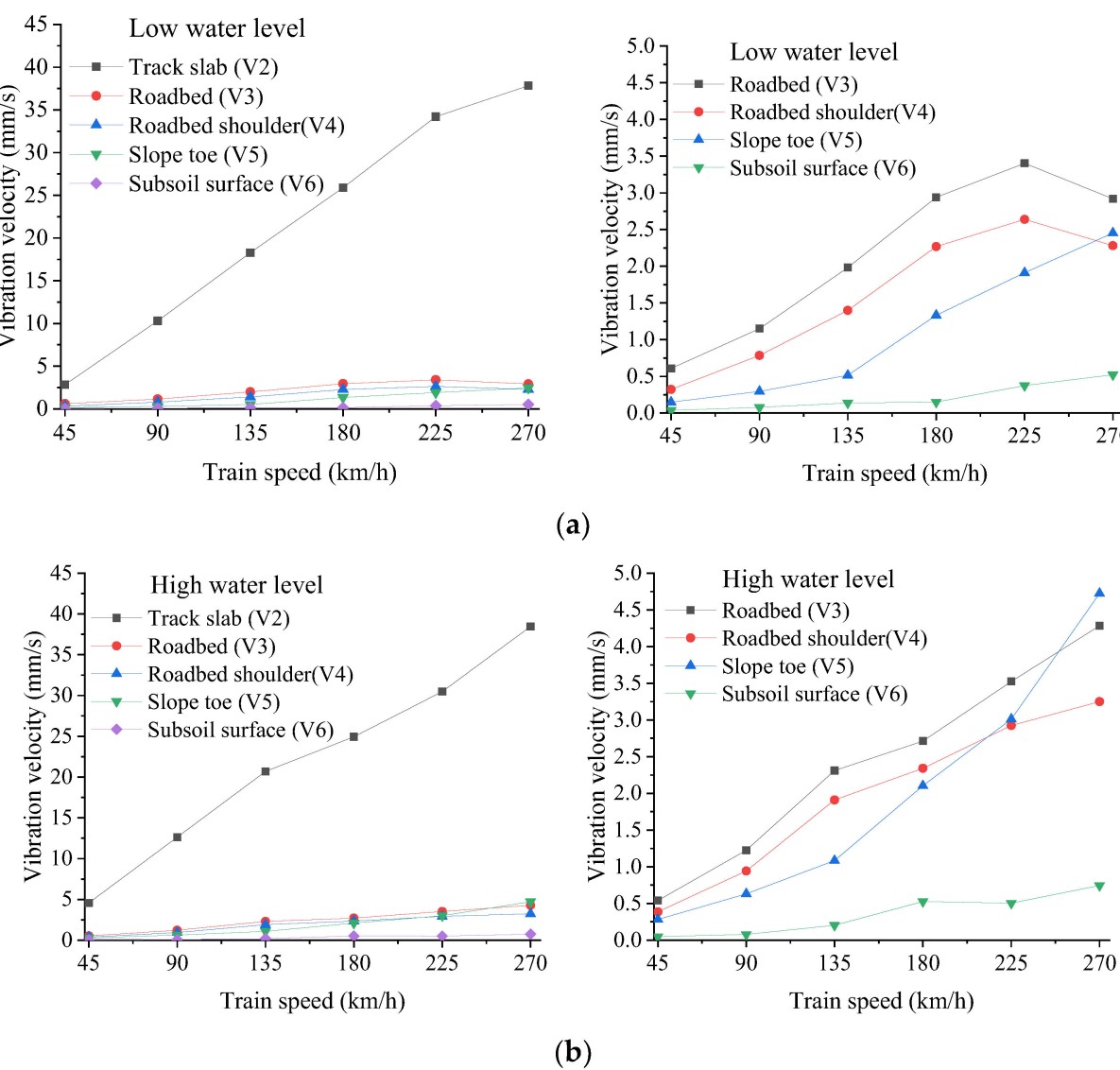

**Figure 16.** Relationship between vibration velocities and train speed at V3 to V6. (**a**) low water level, (**b**) high water level.

Figure 17 shows the relationship between peak vibration velocity and train speed at the locations such as embankment top, raft top, subsoil top, subsoil center, and subsoil bottom. With the increase of train speed, the growth rate and amplitude of peak velocity at V7, V8 are larger than that at V9, V10, V11 in both low and high ground water levels. The peak vibration velocities at various locations of the subsoils are not sensitive to train speed, and the maximum value are lower than 1 mm/s, except for V9 at the train speed of 270 km/h. The piled raft foundation has good vibration isolation for vibration transmission. The dynamic velocities induced by moving train loads have less influence on the vibration response in subsoil.

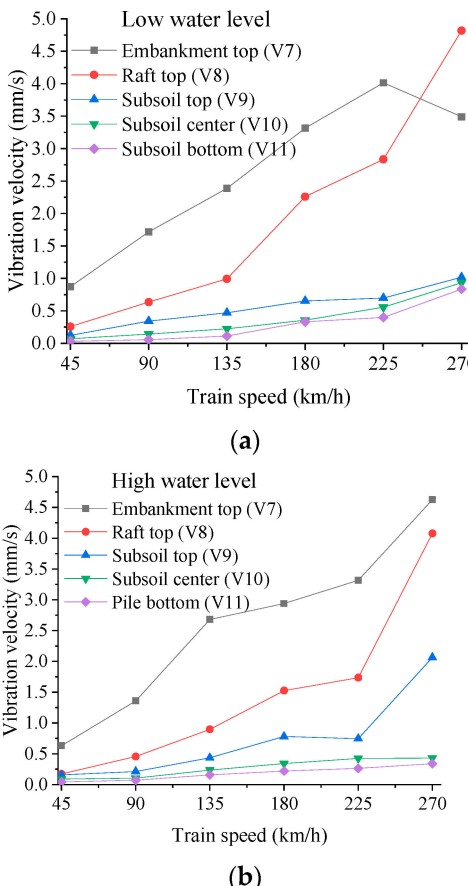

**Figure 17.** Relationship between vibration velocities and train speed at V7 to V11. (**a**) low water level (**b**) high water level.

## 4. Conclusions

An experimental investigation has been carried out into the vibration velocity of track structure, embankment, and piled raft soil foundation using a scaled model testing for trains moving at various speeds in silty soil. The low and high ground water level conditions are considered, of which the comparisons of vibration response are taken out to investigate the influence of water level on the vibration response subjected to train moving loads. A dynamic exciter loading system was used to simulate the dynamic loads on the railway track at various speeds. Based on the operating speed standard, the highest train speed used in model is 270 km/h. Comparisons of vibration velocities at various locations of track structure, embankment, and piled raft foundation have been done in time and frequency domain with different water levels. Some conclusions might be drawn as following:

(1) The time history and peak vibration velocity of the track structure, roadbed, embankment, and piled raft foundation are clearly visible and have sharp impulse and relaxation patterns, corresponding to the loading of train wheels, bogies, and passages. Vibration velocity at the track slab is much stronger than that at roadbed, and sharply decreases when transmitting from track slab to roadbed, reducing by nearly 90%, comes to about 98% at the subsoil surface.

(2) Most of the frequency contents of vibration velocity at various locations are mainly concentrated at harmonic frequencies of 2 Hz, 4 Hz, 6 Hz, 8 Hz, 12 Hz, 18 Hz, 20 Hz for the train speed of 180 km/h, of which the frequency 2 Hz, 6 Hz, and 20 Hz correspond to one carriage length of 25 m, the adjacent bogie spacing of 7.5 m and two wheels spacing of 2.5 m. The change of water level has slight impac on the peak spectrum of vibration velocity at harmonic frequencies.

(3) The vibration velocity levels inside the embankment and subsoil are lower than those on the surface of the track structure and embankment, but still have visible impulse. Vibration velocities decrease quickly in the roadbed and embankment, and then the decreasing rates slow down. With the increase of soil depth, the differences of dynamic velocities in subsoils between different locations become smaller and have very low values.

(4) The dynamic responses of track slab, roadbed, embankment, piled raft, and subsoils are dominated by the dimensions of trains, properties of vibration medium, and load excitation sources. The vibration absorption and attenuation of the embankment and piled raft structure also influence the vibration load transmission and attenuation. The train speeds have more impact on the vibration attenuation in both track structure and substructure.

(5) The vibration velocity attenuations mainly follow the distribution law of exponential curve at different train speed, which can give some empirical guidance for further prediction and analysis on the vibration velocity response of ballastless slab track, embankment, and piled raft supported foundations. The piled raft structure will produce a resistance and excitation effect on the vibration of upper embankment materials.

**Author Contributions:** Conceptualization, Q.F.; Formal analysis, Q.F.; Software, Y.L.; Validation, J.Y.; Writing—original draft, Q.F.; Writing—review & editing, M.G.; Funding acquisition, Q.F., M.G. and J.Y. All authors have read and agreed to the published version of the manuscript.

**Funding:** This study was supported by the National Natural Science Foundation of China (grant number 51908152, grant number 51908150, grant number51908151).

**Institutional Review Board Statement:** Not applicable.

**Informed Consent Statement:** Not applicable.

**Data Availability Statement:** Not applicable.

**Conflicts of Interest:** The authors declare no conflict of interest.

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
