# Peer review of "Experimental Study on Vibration Velocity of Piled Raft Supported Embankment and Foundation for Ballastless High Speed Railway"

_buildings, doi:10.3390/buildings12111982_

Round 1

Reviewer 1 Report

The article presents an experimental investigation about the vibration velocity of track structure, embankment, and piled raft soil foundation using a scaled model testing for train moving at various speeds in silty soil. The testing model was conducted using a reinforced concrete box with dimensions 10 of 5m Length, 4m width, 7m height. Two different underground water levels in soil are considered, of which the comparisons of vibration response are taken out to investigate the influence of water level on vibration response subjected to train moving loads. A dynamic exciter loading system was used to simulate the dynamic loads on the railway track at various speeds. Based on the operating speed standard, the highest train speed used in model is 270km/h. Comparisons of vibration velocities at various locations of track structure, embankment, and piled raft foundation have been done in time and frequency domain with different water levels. The main conclusions are regarding to the time history and peak vibration velocity of track structure, roadbed, embankment, and piled raft foundation, frequency contents of vibration velocity, and the dynamic responses of rail slab track, roadbed, embankment, piled raft, and subsoils.

manuscript's strengths:

The article is well presented with an interesting contribution to the research area;

The introduction is well presented;

The figures and graphs are well presented;

The results were clearly presented and well interpreted by the authors;

The conclusions are well summarized. Overall, I understand that the article is well written. The experimental work will always be interesting for professionals in the field in the context presented in the article;

manuscript's weaknesses:

The abstract needs to be rewritten (the last two sentences are poorly written). Please rewritten the abstract in a short length and include the main conclusions;

Please presents the objectives of the work in a clear and direct way.

The main weakness of the article is the fact that, as it is an experimental work, there are no similar scientific works in the literature. This makes it difficult and impoverished to compare the results obtained in this research with the results previously published and evaluated by other authors.

major recommendations for the improvement of the manuscript:

·      Please rewritten the abstract in a short length and include the main conclusions;

·      A little more in-depth analysis of the results including a comparative analysis of the results with similar works in the literature on the context of the research presented. If there are no similar works in the literature, the authors should make this clear in the work.

Author Response

Changes and response 1: (1) We have rewritten the abstract in a short length and included the main conclusions. Changes have been made using “Track changes” function. (2) We have tried to find similar research results and compared with our experimental results. There are very few studies on the dynamic response of pile raft foundation for high-speed railway in soft soil. Most researches on pile raft foundation of high-speed railway mainly focus on static analysis. But we have tried to make a simple comparison analysis with the Sun’s work [28] on vibration velocity peaks and frequency characteristics of pile raft foundation under train load. Similar frequency response characteristics in vibration velocity are obtained. 1. Changes have been made in red color and shown in 3.1, page 5. For the track slab, roadbed, the peak value of the vibration velocity caused by train wheel load is clearly visible. The time histories of vibration velocity at track slab, roadbed have the variation pattern similar to “M shaped” waves, which correspond to the dynamic loading curves. Sun [28] indicated that the variation pattern of vibration velocity corresponds to loading from the bogies, the time history curves have similar shape of letter “M”. 2. Changes have been made in red color and shown in 3.2, page 10. Bian X [6] compared the amplitudes of the vibration velocities at the track structure and roadbed obtained from the full-scale model testing and the field measurements, indicated that the model testing results have good agreement with the field measurements. In Bian’ study [6], vibrations are found strongest at the track slab and reduce by 60.6%, and 67.5% at the roadbed at train speed of 216km/h, 108km/h. In this paper, compare to reference [6], the vibration velocity has large attenuation rate and amplitude. One of the possible influencing factors may be whether the foundation is reinforced or not by piled raft structure. The above in-depth comparative analysis of the results has been done for the testing position track and subgrade. Further comparative analysis of the vibration response of piled raft foundations has not been carried out for the lack of similar research in related similar works.

Reviewer 2 Report

Fig 2 should be improved, because the quality is low. You can also improve the cleanliness of the figure by separating the texts from the hatches. I am suggesting to use another font (sans-serif).

Can you add additionas figure showing the slab track construction? I have some doubt about the correctness of track the construction. In my opinion the width of the top of the embankement is to large compared with the width od the track slab. Or maybe you simulate the middle track of three track line? But in that case, you should build a larger track slab, which gives different dynamic response. It should be clearly described in the manuscript.

The width (in scale 1:1) of the track slab is 2,5 m. It is to few in my oppinion. One of the most popular slab track construction Rheda 2000 have the width of not less of 2,8 m (for one track), and 3,5 m for the concrete base of the track slab. But the width of the top of the embankement is 8,2 m and it is too much for single track and much so few for three track line.

The slope of the embankment is correct, but it can be specified on the figure 2.

The distance between the track axis and the model testing box is 2,5 m, and it simulates the length of 12,5 m between the track and the building or something else? That distance is extremally small for the HSR. Have you considered the reflected waves, that may have occured?

The remaining drawings are very legible and I have no comments about them.

Author Response

Changes and response 2: (1) Fig.2 has been improved. The missing parts such as waterproof geotextile and foam cotton are added and shown in Fig.2. (2) Considering the reviewer’s questions about the layout of railway track slab structure. Based on the Cord for design of High Speed Railway of China (TB 10621-2014), The slab track used in the model test is China Railway Track System Ⅲ (CRTSⅢ) with width of 2.5m. The scaled model test of ballastless slab track is single track railway, which was designed and developed in Beijing-Shanghai, Wuhan-Guangzhou high-speed railway, and other new railway lines in China. References [6] and [32] both done experimental study on the dynamic behaviour of single track railway, and made correct construction of slab track and embankment. So, different railway tack system type used in high speed railway may have its different construction parameters. Just take the Rheda2000 for example, the width of slab track and concrete base are 2.8m, and 3.5m respectively. In the follow figure, for the CRTS III ballastless slab track system, the width of slab track and concrete base are 2.5m, and 3.1 m respectively. The following changes to the comments are marked in “Track changes” function in 2.1. The track slab of the China Railway Track System III (CRTS III) with a dimension of 4.856m×2.5 m×0.19 m thick was designed, prefabricated in a factory and shipped to the experiment site. A concrete base of dimensions 5×3.1m×0.3m thick was constructed in situ with steel reinforcement. The following references was added in the References of the manuscript as a supplementary. The following picture was used in reference [32] to explain the setup of the experiment model for the convenience of understanding. [32] Zhiping., Zeng; Jundong, Wang.; et al. Experimental study on evolution of mechanical properties of CRTS III ballastless slab track under fatigue load. Construction and Building Materials, 2019, 210,639-649. https://doi.org/10.1016/j.conbuildmat.2019.03.080 (3) Comments: The distance between the track axis and the model testing box is 2,5 m, and it simulates the length of 12,5 m between the track and the building or something else? That distance is extremally small for the HSR. Have you considered the reflected waves, that may have occured? Response: Thanks to the reviewer 's reminder, we have added the missing parts about the waterproof geotextile and foam cotton in Fig.2. The waterproof geotextile and foam cotton can weak the vibration propagation boundary effect of dynamic loading on the wall of model tank. The testing point V6 was 2.49 m away from the track centre. From Fig.13 and Table 3, we can see that the vibration peak velocities at testing point V6 are lower than 2mm/s. Most of the dynamic power was weakened by about 98% when crossing the embankment and soil to V6. For the limitation of this model tank size and scale factor, we make efforts on the attenuations of dynamic velocity in soft soil through material damping and energy dissipation materials. And the waterproof geotextile and foam cotton used in the model test have influence on the vibration wave attenuation. Based on this, the fitting curves of peak velocity were presented, and used to make prediction guidance for position far away from the track centre. As the reviewer mentioned, the experimental result and curves can predict the vibration velocity at building or foundation near the track centre. Researchers can try to enlarge the size of the model boundary in future study, and present numerical simulation to calculate the propagation of vibration velocity, make comparison and verification with model test results.

Reviewer 3 Report

Ref. No.: Manuscript ID: buildings-2000418

Title: Experimental study on vibration velocity of piled raft supported embankment and foundation for ballastless high speed railway

In this paper, a vibration velocity analysis of a large experimental model of a piled raft supported embankment and foundation for ballastless high speed railway was performed. I recommend publication once the comments below are addressed:

1) The authors could start the abstract by contextualizing the problem, starting with something similar to what was stated in the introduction, that is, “as a new reinforcement method in high speed railway lines, the piled raft structure has been used to improve the soil conditions and control settlement”, and then enunciate the paper's proposal. In other words, inform the "why" before the "what" and "how". In this way, it would be clearer for the reader to understand what the paper is about. This is only a suggestion.

2) Verify if sentence “Analysis Results of Basic Characteristics of Vibration Velocity Variation” in the abstract is inserted correctly.

3) There are many errors in citation of references in the introduction. Errors of not inserting "et al"; or when it is inserted, an extra comma is inserted incorrectly; use of the author's first two names; reference [268]; among others. I recommend a general review of the reference citations in whole paper.

4) In the second paragraph of section 2.1, the authors should improve the description of the arrangement of the sensors V7, V7 (V8?), V9, V10 and V11.

5) An explanation of the parameters and scale factors in Table 1 is missing.

6) In Fig. 2 the Vs points of the sensors are not clearly visible. I suggest increasing the font size or putting them in another color.

7) The authors state that the spectral content of the vibration response corresponds to the layout of train carriage, bogies, and wheel loads. However, they do not explain this correspondence of the excitation of the train components with the analytical expression of the force used by the dynamic exciter in the experiments.

8) In this analytical force expression, do the frequencies w, 2w and 3w correspond to the excitation frequencies of the train components?

9) Why was w=40.25 rad/s used for v=180 km/h and w=60.39 rad/s for v=270 km/h? How were the coefficients an and bn of Eq.(1) found?

10) The maximum force value, according to eq (1) is close to 160 kN. If the authors used 5kN, why is the ratio 1:25? Wouldn't it be 1:32?

11) The force functions presented in Fig.3 are not shown completely, their negative part does not appear.

12) Force spectra should be presented for a better analysis of the spectral content of the responses.

13) In section 3.1 it is described that the low and high water level are 5.3 m and 1 m ("respectively" is missing) below subsoil surface. But, in other parts of the paper the low level is informed be 4.3 m.

14) The word “figure” at the beginning of a paragraph should be referred to as “Figure” instead of "Fig.".

15) The beginning of the subsoil surface vibration signal in Fig. 5a shows an initial decay. What is the reason for this? Perhaps this measurement should be re-performed.

16) In Fig. 11 the authors could add a second vertical axis representing the vertical distance from track center.

17) The numbering of the second equation must be corrected. As well as the positioning of the equation on the page.

18) There is no Fig.12 and Table 2 in the paper. Therefore, from pg. 11 all numbering of figures and tables must be corrected.

Author Response

Changes and response 3: (1) The abstract was modified carefully following the reviewer’s suggestion and marked up using the “Track Changes” function. (2) Sentence “Analysis Results of Basic Characteristics of Vibration Velocity Variation” was delete. (3) We have corrected the errors in citation of references in whole paper. (4) The author has improved the description of the arrangement of sensors V7 to V11. (5) An explanation of the parameters and scale factors in Table 1 was shown in 2.1. (6) Figure 2 has been completely modified. (7) The correspondence of the excitation of the train components with the analytical expression of the force used by the dynamic exciter in the experiments was explained clearly in 2.2. (8) The ω=2πf=2πv /Lab corresponds to the excitation frequencies of the train components. The specific explanation contents are shown in 2.2 and marked up using “Track Changes” function. (9) The train speed v=180/3.6=50m/s, Lab =7.8m, ω=2πf=2πv /Lab=40.25 rad/s. The specific explanation contents are shown in 2.2. (10) The F(t)=160kN is used to calculated the corresponding fitting coefficients through equation (1). The simulated axle load of harmonic train in full scale is125kN (12.5t). So the input load magnitude in the exciter is set to 5kN following the size scale factors 1:25. The loading curves of different train axle loads can be presented through the normalized curves at various train speed. (11) For the convenience of dynamic loading, the negative load of dynamic load applied to the track is adjusted and normalized. (12) The force spectra curve is generated and shown in Fig.4 for train speed v=180km/h, and 270km/h. (14) The mentioned problems were completed in whole paper. (15) The reason for initial decay of vibration velocity is a mistake in data processing. The time period for data collection must be consistent, then the time histories of velocities at various have almost consistent fluctuation changes. The right time history curve of vibration velocity at subsoil surface are presented in Fig.5 (a). (16) The vertical distance from testing positions V2, V3, V4, V5, V6 to track center are 0m, 0m, 0m, 0.54m, 0.54m respectively. Additional second vertical axis representing the vertical distance from track center is not necessary. (17) The numbering and positioning of the second equation must was corrected. (18) All the numbering of figures and tables are corrected in whole paper.

Round 2

Reviewer 3 Report

In my opinion, I recommend publication once all my comments in first round were addressed.